# "System Conditions", System Failure, Structural Racism and Anti-Racism in the United Kingdom: Evidence from Education and Beyond

## Paul Miller 

Educational Equity Services, London N1 7GU, UK; admin@educationalequityservices.com

**Abstract:** Racism in any society is fuelled by a number of factors, often acting independently of each other, or, at times, in concert with each other. On the one hand, anti-racism efforts rely on the alignment of four "system conditions" to stand a chance of successfully engaging and tackling racism. On the other hand, where these "system conditions" are not present, or where they are not in sync, this leads to "system failure"—a situation where racism is writ large in society and in the institutions therein, and where anti-racism efforts are severely hampered. Drawing on evidence from within the education sector and elsewhere in UK society, this paper examines how a lack of alignment between "system conditions" hampers antiracism efforts, and simultaneously reinforces racism in society and in institutions—leading to gridlock or "system failure" around anti-racism.

**Keywords:** racism; anti-racism; United Kingdom; "system conditions"; "system failure"

## 1. Introduction

The death of George Floyd in May 2020 has fuelled countless debates about racism and anti-racism in every facet of global society, leading to two important interrelated outcomes. First, these debates have led many organisations, large and small, educational and corporate, public and private, faith-based and non-faith-based, charities and for-profits etc., to consider and examine their complicity in racism, the degree to which their engagement in anti-racism work is deep or shallow [1], and/or the absence of their engagement in anti-racism work. Second, these debates have also led many individuals and organisations to commit to active anti-racism work. The attendant enthusiasm, whether construed as a "push" or a "pull" cannot be denied, and it is clear that the "conscience" of society and organisations has been stirred.

In the United Kingdom (UK), racism is deeply sedimented in the psyche and fibre of all layers of society, and this has been the case for generations [2]. In recent months, I have been describing racism in the UK as a zombie stalking society, for although there are considerable overtures and "statements of commitment" towards tackling institutional and structural racism, there is still a significant segment of society that appear either blind to or unconcerned with racism and its effects. This situation is compounded by the fact that many individuals and organisations appear to lack the understanding and courage to call out racism and the will and power to meaningfully tackle it. This "double lack" is compounded by political correctness among organisational and political leaders that both prevents and softens anti-racist activism as they consider their next move in a vicious game of pacifying; a game in which the spread of racism moves much more quickly and more substantially than attempts to dismantle it.

This paper identifies and examines four "system conditions" that can speed up or slow down the spread of racism and its effects. Where these conditions are not aligned or where they are not in sync, this results in "system failure"—a situation where society is at odds with itself, yielding no clear consensus about what racism is, its impacts, the resources and other interventions required to tackle it, and whose duty (the state, the organisation, the

individual) it is to provide these. What follows is an examination of "system conditions" and their role in anti/racism in the UK.

## 2. Conceptualising Racism and Anti-Racism

The legally adopted definition of racism in the UK is the one offered by Lord MacPherson in the Stephen Lawrence Inquiry Report, which provides that that racism is "Conduct or words or practices which advantage or disadvantage people because of their colour, culture or ethnic origin. In its more subtle form, it is as damaging as in its overt form" [3] (p. 41). From this definition, three important characteristics associated with racism are observable: (i) (it involves) behaviours, speech, practices (individual or organisational), (ii) (it) disadvantages people based on skin colour, culture (including religion, language etc.), ethnic origin (including nationality); (iii) (it) can be overt (e.g., placard bearing, name calling) or covert (e.g., subtle acts to subvert, distort, restrict, gaslight etc.)

Anti-racist/anti-racism is an individual and organisational process of identifying and eliminating racism by changing systems, organisational structures, policies, practices and attitudes, so that power is redistributed and shared equitably [4] From this definition, four key characteristics are observable: (i) (what is it?) an individual and organisational process, (ii) (how does it operate?) identifying and eliminating racism, (iii) (what does it seek to do?) by changing systems, organisational structures, policies, practices and attitudes, (iv) (why does it do this?) so that power is redistributed and shared equitably.

## 3. "System Conditions"

Although this paper does not have space to recall the gamut of events described by [5] (p.206) as "moments in racial time", it is important for us to understand two things. First, structural racism is a feature of historical and contemporary UK society. This "fact of life" is supported by several studies and reports dating back to the late 1940s[1]. Second, moving our understanding beyond the "fact" that structural racism is part and parcel of historical and contemporary British society, to why this is so, provides an opportunity for us to understand the conditions that have either encouraged, propelled or sedimented it.

"System conditions" are structures that explain why a "system" (a nation state or organisations within a nation state) behaves in the way it does. "System conditions" (i) establish the framework within which those delivering relevant services must operate, (ii) shape performance, whether positively or negatively, (iii) improve or undermine processes and (iv) change the way people think about and do their work [6]. In the context of the paper, I have identified four system conditions that can sediment and embed racism in the UK, or which can serve as antidotes to it. I should, however, clarify that for these conditions to be an antidote to racism they must be in sync and be aligned. These four conditions are (i) law and policies, (ii) statements of state representatives and public figures, (iii) national cultural values and attitudes and (iv) funding. To make sense of anti-racism work, we must begin to understand the nature of "system conditions" and how these impact and influence organisations and society. I discuss each of the four system conditions, in turn, drawing on examples from within and beyond the education sector.

### 3.1. The Law, Policies and Official Reports

The law is important for any national society, serving both as a guide and a point of reference for individual and organisational behaviours. On the face of it, the law is the surest and best guarantor of liberation and equity, establishing the parameters of what is "acceptable" within society. The law is often regaled as guaranteeing equal rights and justice for everyone, and jurists are often blamed for their "interpretation" and "application"

---

[1] See for example: (1) Kenneth Little. *Negroes in Britain: A Study of Racial Relations in English Society*, London, Kegan Paul, 1948; (2) James Wickenden. Colour in Britain, Oxford University Press/Institute for Race Relations, 1958; (3) Rampton Report- West Indian Children in Our Schools, 1981; (4) Paul Gilroy. *The Empire Strikes Back: Race and racism in 70s Britain*, London, Hutchinson, 1982; (5) The Stephen Lawrence Inquiry Report/Macpherson report, 1991; (6) The Lammy Review, 2017; (7) Race in the workplace: The McGregor-Smith Review, 2017; (8) Race Disparity Audit, 2017; Paul Miller and Christine Callender. *Race, Education & Educational Leadership in England: An Integrated Analysis,* London, Bloomsbury, 2019.

of the law in cases where injustices are perceived or experienced, although the substantive text or content of the law itself is not usually the subject of debate or blame. Put differently, without the law, there would be conflicts between social groups and communities, although the law itself can be the source of such conflict. Two examples of this may be helpful.

### 3.1.1. Overseas Trained Teachers (OTTs)

Prior to 2008, all teachers recruited to the UK from overseas had to complete a programme of equivalency to be awarded a UK teaching qualification or Qualified Teacher Status (QTS). If a teacher did not achieve QTS within four years of teaching in the UK, they would be required to exit the profession and return home. This is known as the "four-year rule". This was the case for all teachers regardless of country of origin, ethnicity, culture, language, etc. In 2008, with the expansion of the European Union (EU), this changed and teachers from European Union member countries and Switzerland were exempt from the requirements of QTS due to the freedoms guaranteed under the Bologna Convention. Put differently, teachers from EU member countries and Switzerland were no longer subject to the "four-year rule", as their teaching qualification was now deemed equivalent to a UK teaching qualification [7]. However, teachers entering the UK from elsewhere were still subject to this requirement.

In 2012, the law was again changed, and teachers from Australia, Canada, New Zealand and the United States were no longer required to achieve UK QTS as their original teaching qualification was deemed "equivalent" to a UK teaching qualification [7]. However, the requirement for QTS has remained in force for teachers from elsewhere. This change in law/policy has meant that teachers coming to the UK from white-industrialised countries are exempt from doing QTS, but teachers coming to the UK from non-white, non-industrialised countries must continue to do QTS as the qualifications gained in those national systems are not considered to be "equivalent" to a UK teaching qualification [8].

### 3.1.2. Teachers of BAME Heritage

The Equality Act 2010 stipulates that it is illegal to discriminate against anyone with nine protected characteristics in any area of public life, including in recruitment and progression. However, the Equality Act also stipulates that it is illegal to take any of the protected characteristics into account when making decisions about recruitment or progression. As I have discussed elsewhere, " … the Equality Act may be regarded as an instrument for the promotion or encouragement of greater access to opportunities, although offering no [guarantee of] better or different outcomes" and is, therefore, "both as a lock and a key" or "both the motor and handbrake of progress as far as equality practice is concerned" [9] (p. 10). As I have also argued elsewhere, this situation produces a "zero-sum game", where those who need to be most helped by the law are those who appear to benefit the least from its ambit [8].

### 3.1.3. The Sewell Report

In the summer of 2020, Prime Minister Boris Johnson established the Commission on Race and Ethnic Disparities to (i) investigate race and ethnic disparities in the UK, (ii) consider important questions about the current state of race relations in the UK, (iii) understand why so many disparities persist and propose solutions to eliminate or mitigate them [10] (p. 6). The report, published in March 2021, is commonly referred to as the "Sewell Report"—after the Commission's chair, Tony Sewell. Undertaken during a period of global anti-racism protests at structural racism, triggered by the death of George Floyd, the report criticised the "confusing" way in which the term "institutional racism" has been applied [10] (p. 34). Although the report concluded that racism was a "real force" [10] (p. 8) in the UK that needed to be taken seriously, it also concluded that Britain is no longer a country where the system is "deliberately rigged against ethnic minorities" [10] (p. 8) and very few inequalities are directly to do with race.

Part of the report is about how ethnic minorities fare in the UK labour market compared to white people and how disparities in pay, employment, and unemployment have changed over time. The report gives the impression that disparities are falling over time. For example, it refers to an "overall convergence story on employment and pay" [10] (p. 110) but does not present all the available evidence to justify that conclusion. The Sewell Report has been severely criticised as being out of touch with the reality of racism in the UK with hundreds of organisations rejecting its findings, and with hundreds of thousands of people across the UK formally petitioning for its withdrawal. To date, all petitions have been rejected by the government.

### 3.2. Statements of State Representatives and Public Figures

State representatives (e.g., politicians, technocrats, etc.) or public figures (e.g., politicians, journalists, media personalities, etc.) have a significant ability to influence public attitudes and opinion. In the case of "state representatives", although they may not (always) be speaking in the name of, or on behalf of the "state", it may well be assumed by the audience or the public that their views and actions reflect the official state position. Similarly, the opinions of public figures may appeal to a fan base that is unquestioning of their opinions, some of which could be harmful. I offer some examples below.

#### 3.2.1. Kemi Badenoch—Critical Race Theory

During Black History Month (October 2020), Kemi Badenoch, the UK's Equalities Minister, gave a speech in the House of Commons that simultaneously shocked and delighted many people. The target of the minister's speech was critical race theory (CRT). She declared CRT to be a "dangerous trend in race relations . . . an ideology that sees my blackness as victimhood and their whiteness as oppression". She also warned that the Government stands "unequivocally against [it]", and that any school that teaches "elements of critical race theory as fact, or which promotes partisan political views such as defunding the police without offering a balanced treatment of opposing views, is breaking the law". The minister also provided:

> *Our curriculum does not need to be decolonised, for the simple reason that it is not colonised. We should not apologise for the fact that British children primarily study the history of these islands. It goes without saying that the recent fad to decolonise maths, decolonise engineering and decolonise the sciences that we have seen across our universities—to make race the defining principle of what is studied—is not just misguided but actively opposed to the fundamental purpose of education. The curriculum, by its very nature, is limited; there are a finite number of hours to teach any subject. What we have not heard in this debate, from honourable Members on both sides of the House who want more added to it, is what must necessarily be taken out....*

> *Of course black lives matter, but we know that the Black Lives Matter movement is political. I know that because, at the height of the protests, I have been told of white Black Lives Matter protesters calling a black armed police officer guarding Downing Street—I apologise for saying this word—"a pet nigger". That is why we do not endorse that movement on this side of the House. It is a political movement. It would be nice if Opposition Members condemned many of the actions of that political movement, instead of pretending that it is a completely wholesome anti-racist organisation.*

> *Lots of pernicious stuff is being pushed, and we stand against that. We do not want teachers to teach their white pupils about white privilege and inherited racial guilt. Let me be clear that any school that teaches those elements of critical race theory as fact, or that promotes partisan political views such as defunding the police without offering a balanced treatment of opposing views, is breaking the law . . . .*

> *Why does this issue mean so much to me? It is not just because I am a first-generation immigrant. It is because my daughter came home from school this month and said to me, "We're learning Black History Month because every other month is about white history".*

*That is wrong and it is not what our children should be picking up. Those are not the values that I have taught her. They are yet another sign of the pernicious identity politics that look at individuals primarily as groups of biological characteristics. People often do not realise when that has taken hold, and I know that many of them are well meaning . . . . History tells us that this is a country that welcomes people, and that black people from all over the world have found this to be a great and welcoming country* [11].

The minister's speech shows three things. First, it is an excellent example of a state actor fronting a view that may or may not be the official position of the state, but through their position and status have offered a view that could force many persons thinking about doing anti-racism work to do a double-take, or those already involved in anti-racism work, into a lock jam. Second, such a speech would have delighted individuals that do not believe that racism exists in the UK. Third, it shows a lack of understanding of critical race theory (CRT), which assumes a stance of "non neutrality". That is, CRT refutes being "non racist" as an acceptable position, opting instead to promote a position of being "anti-racist" which is an active stance against racism and not merely a declaration.

### 3.2.2. Liz Truss—The New Fight for Fairness

In a speech to the conservative think-tank Centre for Policy Studies, two months after Kemi Badenoch's speech, Liz Truss, Minister for Women and Equalities, proposed sweeping changes to current equality legislation. She criticised the Equality Act 2010 and dismissed unconscious bias training as one of several "tools of the left" that do nothing to "fix systems". She stated that the state's agenda had become too narrow, and the discrimination debate should not focus solely on race, religion, sexual orientation and disability.

The minister criticised equality legislation, claiming that it "overlooks socio-economic status and geographic inequality". She insisted the debate surrounding equality must be "led by facts . . . not by fashion" and emphasised the need for objective data. Significantly, she argued that data based solely on protected characteristics—over which it is illegal to discriminate—age, disability, gender reassignment, marriage and civil partnership, pregnancy and maternity, race, religion or belief, sex, and sexual orientation—were not fit for purpose when it comes to setting equality policy. She also said:

"Underlying this [approach] is the soft bigotry of low expectations, where people from certain backgrounds are never expected or considered able enough to reach high standards. This diminishes individual humanity and dignity, because when you choose on the basis of protected characteristics, you end up excluding people" [12].

The government, she said, would now turn to promoting "freedom, choice opportunity, individual humanity and dignity . . . " and that equality discussions had "been dominated by a small number of unrepresentative voices, and by those who believe people are defined by their protected characteristic and not by their individual character". Truss further said "Techniques like unconscious bias training, quotas and diversity statements . . . do nothing to make the workplace fundamentally fairer. By driving reforms that increase competition, boost transparency and improve choice, we can open up opportunities". She warned that the state must not "waste time on misguided, wrong-headed and ultimately destructive ideas that take agency away from people". She advised that the Conservative Party's approach to equality would "reject the approach taken by the Left . . . captured as they are by identity politics, loud lobby groups and the idea of 'lived experience'" and would instead " . . . focus fiercely on fixing geographic inequality . . . addressing the real problems people face in their everyday lives . . . using evidence and data" [12].

The Minister's speech is another signal from the (Conservative) government that it does not believe unconscious bias training can raise consciousness about racism and other forms of discrimination. Since 2015, civil service managers had been required to complete mandatory unconscious bias training in an attempt to tackle implicit and explicit biases and build more inclusive workplaces. However, Cabinet Secretary Julia Lopez was reported as saying such training would be phased out by government departments, and public sector organisations would be encouraged to do the same [13]. It has also been widely reported

that many Conservative MPs and senior figures were not happy with the influence the Black Lives Matter movement has exerted since summer 2020, and the minister's speech could be viewed as a form of "fight back".

### 3.2.3. Laurence Fox—The Duchess of Sussex and Sikhs

During an appearance on the BBC's Question Time, British actor, Laurence Fox, claimed it was "racist" for an audience member to refer to him as a "white privileged male" [14]. The actor also refuted that the treatment of Meghan Markle, the Duchess of Sussex, by the British press was racist. Clashing with audience member Rachel Boyle, a university lecturer and race and ethnicity researcher, who suggested the way Meghan Markle had been treated in the press was "racist", Fox responded by saying, "It's not racism, we're the most tolerant, lovely country in Europe. It's so easy to throw the charge of racism at everybody and it's really starting to get boring now". She responded, "Says a white privileged man" to which Fox remarked, "I can't help what I am, I was born like this, it's an immutable characteristic, so to call me a white privileged male is to be racist—you're being racist" [15]. Footage of Fox's appearance was widely shared on social media—with some praising his comments but others calling them offensive. Later, writing on Twitter, Fox said, "Still waiting for a single example of anything I've ever said or done that could ever be deemed racist" [15].

This is not the only time Fox has made unsavoury comments about individuals of BAME heritage. The actor had previously referred to "the oddness in the casting" of a Sikh soldier in Sir Sam Mendes' movie 1917. To amplify his point, he argued:

"It is kind of racist—if you talk about institutional racism, which is what everyone loves to go on about, which I'm not a believer in, there is something institutionally racist about forcing diversity on people in that way. You don't want to think about [that]" [14].

Fox's comments drew widespread criticism and historians clarified that about 130,000 Sikh men fought in the British Army during World War One. This number was circa 20% of the British Indian Army according to the WW1 Sikh Memorial Fund. Fox later tweeted an apology, stating:

"Fellow humans who are Sikhs, I am as moved by the sacrifices your relatives made as I am by the loss of all those who die in war, whatever creed or colour. Please accept my apology for being clumsy in the way I expressed myself" [14].

Fox's comments are consistent with views held by other public figures and members of the wider populace who may see him as a "mouthpiece".

### 3.2.4. Eamon Holmes—The Duchess of Sussex

In November 2019, broadcaster and journalist Eamonn Holmes called the Duchess of Sussex "uppity", a word which was used as an insult to black slaves in the USA who their white slave owners thought "didn't know their place". The Duchess had demanded that the press should respect her privacy. Responding to complaints made by the public about Holmes' comments, ITV's Head of Diversity Ade Rawcliffe, provided the following:

At the time of the broadcast in July, Eamonn Holmes was unaware of the history of the particular usage of the term "uppity" and how it could have been interpreted when describing Meghan Markle. Eamonn was using the term to describe what he interpreted as arrogance [16].

In a radio interview on "talkRADIO" in January 2020, Holmes said of the Duchess of Sussex,

"I just find her incredibly irritating. I've never met her, but I look at her and I think 'I don't think I'd like you in real life'. Awful, weak, woke, manipulative, spoilt . . . " [17].

Holmes' recent 'attack' on the Duchess of Sussex, the first member of the UK's Royal Family of mixed heritage, undermines the earlier apology and explanation put forward by ITV on his behalf some months earlier, and arguably captures the opinions of millions of people in the UK with similarly racist views.

### 3.2.5. Boris Johnson—Black People and Muslims

In a Spectator column written in 2002, and referencing a visit to Africa by the then Prime Minister Tony Blair, Johnson said "What a relief it must be for Blair to get out of England. It is said that the Queen has come to love the Commonwealth, partly because it supplies her with regular cheering crowds of flag-waving piccaninnies". In the same column Johnson described African people as having "watermelon smiles" and observed that British colonialism in Africa is "not a blot upon our conscience", adding "The problem is not that we were once in charge, but that we are not in charge anymore" [18].

Questioned about his comments during his first campaign for London Mayor, Johnson claimed his comments had been "taken out of context". Three years later, in 2005, Johnson wrote in the Spectator that he believed it was only "natural" for the public to be scared of Islam. He suggested:

"To any non-Muslim reader of the Koran, Islamophobia—fear of Islam—seems a natural reaction, and, indeed, exactly what that text is intended to provoke. Judged purely on its scripture—to say nothing of what is preached in the mosques—it is the most viciously sectarian of all religions in its heartlessness towards unbelievers" [19].

Following the 7 July 2005 London bombings (7/7), Johnson also questioned the loyalty of British Muslims and insisted that the country must accept that "Islam is the problem. It will take a huge effort of courage and skill to win round the many thousands of British Muslims who are in a similar state of alienation, and to make them see that their faith must be compatible with British values and with loyalty to Britain. That means disposing of the first taboo, and accepting that the problem is Islam. Islam is the problem" [19].

He questioned, "What is going on in these mosques and madrasas? When is someone going to get 18th century on Islam's medieval ass?" [19].

In 2018, as Foreign Secretary in Theresa May's government, Johnson was reported to the Equalities Commission after comparing Muslim women who wear burqas to "letter boxes" and bank robbers. Johnson also provided, "it is absolutely ridiculous that people should choose to go around looking like letter boxes"; adding that any female student who appeared at school or in a lecture "looking like a bank robber" should be asked to remove it [20,21].

The comments by Johnson, made before he became Prime Minister, illustrate racist attitudes towards Black Africans and Muslims. His "othering" of these groups is consistent with the UK's officially adopted definition of racism, which includes "Conduct or words or practices which advantage or disadvantage people because of their colour, culture or ethnic origin.... [3] (p. 41).

### 3.3. National Cultural Values and Attitudes

A society sets the values and standards of behaviour it believes to be acceptable. Two primary functions of laws and policies, for example, are the creation and maintenance of the social order. That is, laws and policies exist to maintain order and coordinate behaviour and transactions among citizens. This aspect of the rule of law holds that a framework of legal or other rules governs social behaviour, and that individuals must generally behave in ways that maintain or do not breach these. The introduction of the "Prevent duty" and "Fundamental British Values" are illustrations of this.

In November 2014, the Conservative—Liberal Democrat coalition government directed that schools must promote British values. It could be argued that prior to the government's introduction of what it called "Fundamental British Values" (FBV), the values held by individuals and groups in Britain were implicit, and the articulation of these values was an attempt to make explicit and common a set of values to which all citizens could subscribe. According to Ofsted [22] "fundamental British values" are:

- democracy;
- the rule of law;
- individual liberty;
- mutual respect;

- tolerance of those with different faiths and beliefs and for those without faith.

The introduction of FBV has been met with significant criticism and scepticism due to its connection with the controversial "Prevent Strategy" [23]. The stated aim of the Prevent Strategy is to "reduce the threat to the UK from terrorism by stopping people becoming terrorists or supporting terrorism", with its three primary objectives being:

- "respond to the ideological challenge of terrorism and the threat faced from those who promote it";
- "prevent people from being drawn into terrorism and ensure that they are given appropriate advice and support";
- "work with sectors and institutions where there are risks of radicalisation"

The government states that the Prevent duty is designed to:

"Deal with all forms of terrorism and with non-violent extremism, which can create an atmosphere conducive to terrorism and can popularise views that terrorists then exploit" [23] (p. 6).

The government defined "extremism" in this context as:

"Vocal or active opposition to fundamental British values, including democracy, the rule of law, individual liberty and mutual respect and tolerance of different faiths and beliefs" [23] (p. 107).

In addition, the somewhat problematic articulation of so-called "British values" means that several individuals and groups may have, at some point, contravened these.

In a press release on 27 November 2014, the Department for Education told all schools to promote "British values" and that Ofsted wanted to see a school ethos and climate that promotes "British values" at every level [24]. Ofsted later confirmed it would assess "British values" through spiritual, moral, social and cultural development (SMSC), the curriculum and school leadership [22].

As mentioned earlier, there has been strong criticism and scepticism about the introduction of Prevent and FBV because these were regarded as divisive, as the Prevent Strategy "disproportionately targets British Muslims as a 'suspect community'", as well as securitising society and undermining community cohesion" [25] (p. 1). According to Hechter and Horne [26], the social order is maintained by domination, through the power of those with the greatest political, economic, and social resources. Thus, the Prevent Strategy has come to be regarded as a mechanism for stoking racism, imposed by the ruling class, and not something co-constructed with society for the "common good". The same could be said of the FBVs, which were not co-constructed with society as a whole, and which did not include key values such as "equity" (the fair treatment, access, opportunity, and advancement for all people) and "justice" (fairness: a system of law in which every person receives their due, including all rights, both natural and legal) which many people seek, and which many appear to lack in Britain, including those of BAME heritage.

Another example of how "society sets the standards for itself" is the recent and ongoing racist abuse meted out to British footballers of BAME heritage who "take the knee" before matches. Despite condemnation, and fans being banned from matches, racist abuse has become commonplace. Globally, taking the knee has come to symbolise an act of solidarity against racism among athletes of all ethnic origins. Nevertheless, fans and commentators continue to chant names at and single out players and athletes of BAME heritage for racist abuse—a clear sign they want no part in or do not believe in anti-racism. In December 2020, Cabinet Minister George Eustice failed to condemn Millwall football supporters who booed players taking the knee [27].

Furthermore, in December 2020, Sainsbury's Supermarket decided to air adverts featuring Black families which led to accusations of virtue signalling, a flurry of racist comments on social media, and threats to boycott the supermarket. In response, Sainsbury's released a statement reaffirming its commitment to being an inclusive retailer, with other supermarkets Aldi, Asda, Co-op, Iceland, Lidl, Marks & Spencer, Tesco and Waitrose showing support, with the hashtag #StandAgainstRacism.

On top of this, The Times had previously reported in September 2020 that 40 Conservative MPs had refused to undertake unconscious bias training, challenging whether unconscious bias exists [28]. These examples are compounded by the fact that the Teacher Diversity Steering Group, a volunteer group of researchers, school leaders and other experts in equity, diversity and inclusion (EDI), which met quarterly at the DfE's offices in London, and which was first convened during Theresa May's premiership shortly after the publication of the 2018 Race Disparity Audit, has never met since Boris Johnson took office as Prime Minister. There have also been several examples within the last two years of several students of BAME heritage at UK universities being racially abused [29]—affirming society's role at multiple levels in promoting racism instead of advocating against it.

This is not to suggest that all people in a society will have the same beliefs or values, but rather that if the values of a society are defined by and subscribed to by all of society, then members of that society will show an inclination to these values and to each other. However, where values are imposed by the political class, and where values are not co-constructed by citizens, and when social interventions appear to exploit tensions, anxieties and fears among ethnic groups, and individuals based on ethnic, cultural, linguistic or other ascriptions, this can only make the work of anti-racism much more difficult.

*3.4. Funding*

Racism is costly, and detrimental to individuals, families, ethnic groups and society as a whole. Yet, it is more costly to ignore the perils and impacts of racism by not doing anything to stop its spread and mitigate its impacts. Consequently, anti-racism efforts require significant financial investment for consulting and action planning, programmes and activities, training and behaviour change, and processes and systems change. One example from education may help to clarify.

In 2014, the Department for Education (DfE) established the Leadership for Equality and Diversity Fund (LEDF) with the primary aim of supporting schools to develop solutions to help teachers of protected characteristics progress into leadership [30]. This was in recognition that diversity in the teacher workforce was problematic and was even more problematic at higher levels within the progression [9].

The introduction of the LEDF was a manifestation of the DfE's "Statement of intent" on the diversity of the teaching workforce in which it highlighted women and ethnic minority teachers were under-represented at senior levels, pledging it wanted to see "a teaching profession that prides itself on promoting a diverse workforce" [31].

Under the LEDF, eight regional hub lead schools would allocate funding to school-led projects throughout their area—a situation in which the DfE argued that the funding "reinforces the government's commitment to increasing the diversity of school leadership and maximising the number of leaders available by raising aspirations and the chances of successful promotion among people with leadership potential" [30]. Between 2018 and 2020, the DfE had invested £2 million into the diversity hubs.

Funding for projects in 2019/2020 was delayed due to the Covid-19 pandemic, however, in December 2020, the DfE announced that the funding would be scrapped altogether. Writing to hub leaders, the DfE states "Current programmes under the Equality and Diversity Fund, delayed from the summer as a result of Covid-19 will end in December 2020 and beyond this, DfE has taken the decision not to proceed with a further round of E&D Hubs" [32]. The DfE also noted it "continually reviews programmes to ensure they continue to address sector needs" and that it was "currently exploring other strategies for supporting teacher workforce development" [32]. The withdrawal of the LEDF has been met with strong condemnation from sector leaders, and equity, diversity and inclusion advocates.

Without (adequate) financial investment to attend to a range of interventions, anti-racism work could remain largely philosophical and shallow. Financial resourcing represents a catch twenty-two scenario where funding for a range of initiatives and interventions is vital to successfully confronting and addressing deep-seated structural, institutional and

personal attitudes around "race" and racism, and where the absence of funding starves schools and other institutions of much needed financial oxygen to leverage the best advice and interventions possible, and to invest in a combination of context-relevant initiatives and solutions.

### 4. System Conditions, Structural Racism and Anti-Racism

Former Jamaican Prime Minister, P.J. Patterson is widely reported as saying "the law is not a shackle, but rather a tool for social engineering". Underpinning this view is an assumption that those who create the law, and those who apply or enforce the law, should do so in a "positive spirit" or in ways that lead to positive outcomes for society. This assumption positions the law and attendant policies as positive tools in societal transformation. However, this assumption may not be shared by all, including those who are responsible for creating the law and attendant policies in the first place. In the case of overseas trained teachers (OTTs) from non-white, non-industrialised countries, it appears existing UK law and policy, as conceived and applied, regarding teacher migration and qualification equivalency, are directed towards the creation of a two-tier system where those from white industrialised countries are advantaged and where those from non-white, non-industrialised countries are disempowered and disadvantaged [9]. This is a foundational element of structural racism.

In the 1950s–1960s, people from Africa, the Caribbean, and the Indian sub-continent were actively sought out to come to the UK to assist with rebuilding efforts in the immediate decades following World War II. These are the people who collectively became known as the "Windrush generation", the majority of whom were skilled technicians and tradespeople. In the 1970s, things had improved for the country as a whole and in response the government curtailed its previous "open arms" approach to migrants from Africa, the Caribbean and the Indian sub-continent, instead prioritising migrants based on "educational skills and resources" [33] (p. 19). Many, however, saw this as a direct measure to limit migration to the UK from the Indian sub-continent, Africa and the Caribbean—especially because many countries in these regions were themselves trying to grow their own human capital base, having only recently won their independence from colonial Britain. Nevertheless, this paved the way for many educated and qualified people to come to the UK from these regions to work, especially in areas such as nursing and teaching.

Notwithstanding, UK law regarding overseas trained teachers took on its most racialised and insidious form in 2012 when legislators approved that only teachers from non-white, non-industrialised countries (e.g., Jamaica, South Africa, India, etc.) arriving in the UK would still need to undergo compulsory training and assessment for UK Qualified Teacher Status [34]. Put differently, for teachers from non-white non-industrialised countries arriving in the UK, their country of origin and their qualifications (cultural capital) became the new "head tax" for entry and progression in their career in Britain—a situation that produces the same effect as if they were restricted because of the colour of their skin as the teachers arriving from non-white non-industrialised countries to the UK are not white [9].

The Prevent Strategy arguably has a similarly anti-positive effect which, instead of promoting and building community and social cohesion, has created and exposed deep fractures and a severe lack of trust between the white majority, in particular middle and upper classes, and those from BAME heritage, in particular Muslims. It is curious in both cases how the law might be seen as a tool for "societal transformation" and not a tool to "shackle" and disadvantage members of a society. Hegemony promotes (and sometimes requires) pacification and disempowerment of peoples and cultures through regulatory frameworks allegedly, in the interest of a common good [35]. But what is this common good, and whose is this common good? The treatment of overseas trained teachers from non-white non-industrialised countries and Muslims gives rise to the view that hegemony enlists the help of disempowered in their own disempowerment [35,36], amplifying a

"cloak of whiteness"—structures that re/produce white privilege, power and domination of non-whites [37,38].

That the Equality Act 2010 promotes equal opportunities of access but does not guarantee equal opportunity of outcome is another example of UK legislation that has not led to any demonstrably different outcomes as far as "race" equity is concerned; while it makes it illegal to deny someone employment because of "race"/ethnicity, it also makes it illegal for "race"/ethnicity to be taken into consideration when making recruitment and/or progression decisions. This produces a zero-sum game, where the rights and privileges promised under the law are not guaranteed [8].

A national society sets the standards by which members of that society will be judged, and by which members of that society can judge it. Although it is much easier for a society to impose judgments and limits on its members, a society may not readily consider how the rules, laws, policies, cultural and other practices can and do shape behaviour and attitudes. The laws and policies introduced by politicians and/or governing parties are, therefore, helpful in our understanding of their take on issues, for example, "race" equity. Furthermore, these laws and policies are vital to our understanding of the equity fibre and landscape, as well as its development. Put differently, if state representatives and public figures themselves hold racist views, if they engage in racist labelling, if they deny racism exists, if they refuse to call out or challenge racist incidents, and if they are anti anti-racism, then this has severe implications for the moral and social fibre of society.

On top of this, if state representatives appear numb to racism, then the laws and policies they promote and lobby for will not promote "race" equity, but rather a form of "colour blindness" at best, and post-racialism at worse. Further, if state representatives and public figures are tone-deaf to racism and its marginalising effects, they will more likely promote messages and systems that undergird this numbness and deafness, which not only risks undermining the work of anti-racists, but simultaneously risks legitimating racism as an acceptable practice and way of life among their followers. In both cases, the risks and realities of systemic racism are not only ignored but reinforced. Numbness to racism by state representatives and public figures only begets numbness to racism within society, and policies, practices, systems, values, behaviours and attitudes, individuals and organisations will normalise racism in everyday interactions, thereby keeping it firmly in place, instead of dismantling it.

Dismantling racism requires significant financial investment, and so does creating pathways for "levelling up" BAME staff in organisations. These two things go hand in hand, and the creation of pathways to level up is a fruit of dismantling systems, processes and policies that are restrictive or do not work. Levelling up, therefore, requires training, expert consultancy support and/or advice, institutional focused audits, studies and culture reviews, all of which require financial investments. As [9] (p. 1003) noted, "doing race equality is serious business, and doing race equality in schools or educational institutions is serious business that requires courage and the moral use of power that extends beyond sympathising to taking actions". Figure 1 shows the four "system conditions" when they are aligned.

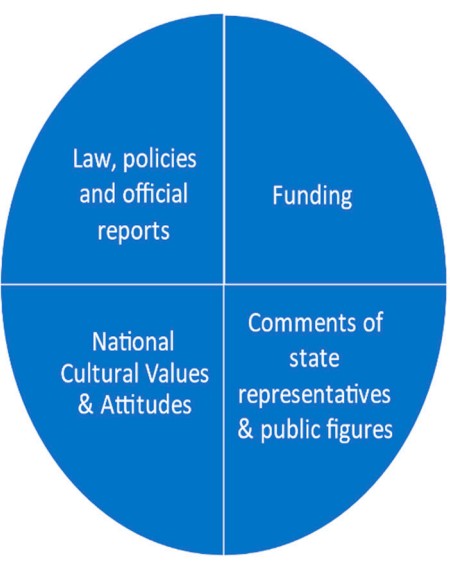

**Figure 1.** Anti/racism "system conditions".

## 5. System Conditions, System Failure

System failure has been variously defined. On the whole, however, it is regarded as "any material failure, fault or problem of the systems" or "one or more malfunctions [which] results in pollution . . . , contamination . . . ., nuisance . . . or a hazard to public health" [39]. The causes of system failure are multiple and varied, and have variously included defect or non-functional requirements, poor (development) practices, incorrect assumptions regarding system requirements, poor user interface, inadequate user training/user error, and poor fit between systems and organisation. As also noted by [39], system failure occurs when a system does not meet its requirements. Other causes of system failure include ineptitude in the management of diversity, institutional pathology and structural variables that concentrate power centrally. In the context of this paper, I have identified four "system conditions" that can work successfully to tackle racism, or which, if not aligned, or if absent, can and will lead to system failure. That the law shackles some but disadvantages others, is problematic. That the law creates mistrust and suspicion is problematic. That the law promises equity but does not guarantee this is problematic. That state actors and public figures can spew racism and stoke racist tensions under the guise of parliamentary privilege or "free speech" is problematic. That the government has withdrawn funding to tackle underrepresentation in leadership in schools is problematic.

Crucially [40] argues that all other systems converge on political systems, arguing "Political system is that part of society engaged in authoritative allocation of values" (p. 52). Central to this system are institutions, structures and defined allocative mechanisms that work together to shape a national environment (culture, socio-economic conditions etc.) through the character and content of policies, and other interventions. Thus, as a system condition, the political system is the most important element promoting and deciding the law, in the shaping of values, and in the allocation of resources.

It is important to understand that for the system conditions to be successful in tackling racism, each must be present and they all must be in sync. Based on the available evidence provided in this paper, and based on wider evidence across society (too much to be included herein), it is not difficult to conclude that regarding anti-racism in the UK, society and its organisations have reached a crisis point that amounts to a system failure. Where system failure has occurred, there is likely to be disharmony and suspicion between and among communities, suspicion of the state and lack of trust in the authority of the state and/or its officials [41]. Thus, system failure not only sediments existing problems within a system but depletes the capacity of the state and members of civil society to resolve the tensions therein. System failure manifests itself when:

- The political system adopts a different set of values and expectations to institutions;
- There are clashes in goals pursued within the system;
- Constructive forces of change meet conservative forces against change;
- The system continues to promote strong kinship and ethnic affinities and disregard the larger umbrella system [42] (p. 14).

Evidence of all these elements are present in the UK, with each contributing to the embedding of racism in society. Figure 2 shows the four "system conditions" out of alignment, implying a failure of the system as a whole.

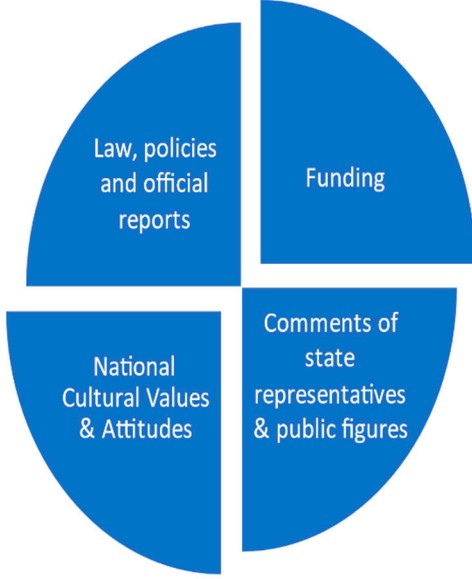

**Figure 2.** Anti/racism "system conditions" and "system failure".

Racism in any form is unacceptable, and current levels of public support for and displays of racism, and the current wave of anti anti-racism in the UK means that racism is a public hazard, and that BAME individuals and groups are "at heightened risk" of further individual, institutional and systemic discrimination.

## 6. Concluding Thoughts

In the UK, if politicians and the wider populace wants to seriously tackle deeply embedded structural racism, then "system conditions" must reflect this. "System conditions" are potent shapers of national cultural and organisational practice, and, thus, the work of dismantling structural racism in the UK will only begin to bear fruit across all segments of society and within our organisations when these conditions are in place and in sync. Political leaders should craft and implement policies and laws that are anti-racist in intent and scope, and their actions and words should also align with these, seeking to promote and galvanise social cohesion.

If the law promotes but does not guarantee equity, and if those with the legal authority, as provided by the electorate, to develop the law do so in a manner to disadvantage some citizens, instead of to liberate all citizens, then those tasked with interpreting and applying the law and policies are at acute risk of being complicit in racism. The unfair policy treatment in the form of racialised knowledge [8] is important in our understanding of how the law can be used to create advantages for some and not all, and the Prevent Strategy is widely regarded as a socially divisive, if not anti-Muslim invention.

For the last five to six decades, despite significant challenges to racism from anti-racists, there has also been an equally perilous push by very powerful forces and elements of British society to silence those who speak out against racism, and to get the already marginalised to be complicit in their own marginalisation. These issues should not be over-simplified, and often the strongest leaders recognise that they do not have all the answers,

but they demonstrate a personal and an organisational commitment to anti-racist activism. Although not everyone in an organisation or indeed in society may accept that racism exists in the UK and that it needs to be tackled, it is crucial to also note that without a collective conscience and shared values, beliefs and attitudes around anti-racism, achieving social order is nigh impossible and social order is crucial for the wellbeing of society. In such a situation, the failure of the system is once more highlighted, requiring urgent political and organisational leadership, financial investment and public scrutiny, and for racism to be classified as a safeguarding issue.

**Funding:** This research received no external funding.

**Institutional Review Board Statement:** Not applicable.

**Informed Consent Statement:** Not applicable.

**Data Availability Statement:** Not applicable.

**Conflicts of Interest:** The authors declare no conflict of interest.

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
