# Peer review of "“System Conditions”, System Failure, Structural Racism and Anti-Racism in the United Kingdom: Evidence from Education and Beyond"

_societies, doi:10.3390/soc11020042_

Round 1
Reviewer 1 Report
The paper engages with a very important and interesting topic by discussing the concepts of racism and anti-racism in relation with the systemic social, political, and financial conditions affecting the perception and the understanding of this social phenomenon. These conditions include: a) the law and policies concerning anti-racism and both social and workplace discrimination, b) the public statements of state representatives and public figures regarding anti-racist policies and measures, c) the national cultural values and attitudes, and d) funding.
The author places her/his analysis in the broader context of the UK society and argues that “where these factors do not align… or where one or more factors is missing, this leads to ‘system failure’ or a condition where racism is writ large in society, and where anti-racism efforts are severely hampered” (Abstract, lines 11–14). The author concludes by arguing that both the past and the recent social and political experience in the UK society points to this direction and that there is an urgent need to rethink the structural conditions of anti-racism.
From a methodological point of view, I fully agree with the author when she/he tries to defend a global sociological perspective by reconstructing the conditions shaping the effective contestation of racism on both the social, the political/financial, and the educational levels. The overall analysis is good and convincing. Any account of racism should take seriously into account the social and political contexts where law, social values, and normative expectations are developed, recognized, and eventually contested. This is a well-written article, which provides a vast amount of empirical evidence useful for the social scientist in order to corroborate the development of its main argument from an empirical point of view.
Some minor problems: a) There is virtually no definition – at least a working one – or even a brief account of what racism theoretically means. A brief discussion of the concept of racism is of primary importance in order to fully accommodate the importance of the multiple perspectives related to it both conceptually and empirically. The author should complement this analysis with some references related to the concept of racism; b) the author does not explain sufficiently enough why she/he considers structural racism an ineradicable element of UK society (lines 57–58). She/he refers to “several studies and incidents”, but the references are missing.
Author Response
Reviewer 1
The paper engages with a very important and interesting topic by discussing the concepts of racism and anti-racism in relation with the systemic social, political, and financial conditions affecting the perception and the understanding of this social phenomenon. These conditions include: a) the law and policies concerning anti-racism and both social and workplace discrimination, b) the public statements of state representatives and public figures regarding anti-racist policies and measures, c) the national cultural values and attitudes, and d) funding.
The author places her/his analysis in the broader context of the UK society and argues that “where these factors do not align… or where one or more factors is missing, this leads to ‘system failure’ or a condition where racism is writ large in society, and where anti-racism efforts are severely hampered” (Abstract, lines 11–14). The author concludes by arguing that both the past and the recent social and political experience in the UK society points to this direction and that there is an urgent need to rethink the structural conditions of anti-racism.
From a methodological point of view, I fully agree with the author when she/he tries to defend a global sociological perspective by reconstructing the conditions shaping the effective contestation of racism on both the social, the political/financial, and the educational levels. The overall analysis is good and convincing. Any account of racism should take seriously into account the social and political contexts where law, social values, and normative expectations are developed, recognized, and eventually contested. This is a well-written article, which provides a vast amount of empirical evidence useful for the social scientist in order to corroborate the development of its main argument from an empirical point of view.
Some minor problems: a) There is virtually no definition – at least a working one – or even a brief account of what racism theoretically means. A brief discussion of the concept of racism is of primary importance in order to fully accommodate the importance of the multiple perspectives related to it both conceptually and empirically. The author should complement this analysis with some references related to the concept of racism; b) the author does not explain sufficiently enough why she/he considers structural racism an ineradicable element of UK society (lines 57–58). She/he refers to “several studies and incidents”, but the references are missing.
I have now added a footnote where I have signalled a number of studies and reports.
It’s not that racism is ineradicable, but rather that racism will only be eradicated when/ where the system conditions are in place and in sync. This was previously set out in the conclusions, but I have now strengthened this in the abstract also.
Reviewer 2 Report
The conclusions made by the author are not surprising. The most interesting aspect is the discussion of the requirements enforced on teachers and the inequality in those requirements, which is drawn from previous research undertaken by the author. What would be helpful with this paper would be for the author to conclude with an example of an alignment of anti-racist systems where there was not system failure. This example will probably have taken place at a local level rather than the national level that is the subject of much of this essay.
Overall the writing is good.
Author Response
The conclusions made by the author are not surprising. The most interesting aspect is the discussion of the requirements enforced on teachers and the inequality in those requirements, which is drawn from previous research undertaken by the author. What would be helpful with this paper would be for the author to conclude with an example of an alignment of anti-racist systems where there was not system failure. This example will probably have taken place at a local level rather than the national level that is the subject of much of this essay.
I agree 100% it would be good to find a positive example, but currently there are no examples of the all four ‘system conditions’ being aligned or being in sync that I can point to.
Hopefully the paper will be helpful to show politicians, educational leaders and the general population what is required to improve conditions.